# Cultivation of Microalgae *Chlorella vulgaris* in Open Reactor for Bioethanol Production

**Graziella Silva** [1,†], **Keilla Cerqueira** [2,†], **Jacqueline Rodrigues** [1,†], **Karollyna Silva** [1,†], **Diego Coelho** [3,*,†] **and Roberto Souza** [1,†]

1  LABAM—Laboratory of Environmental Biotechnology, Federal University of Sergipe. Av. Marechal Rondon, s/n, Jd. Rosa Elze, São Cristóvão 49100-000, SE, Brazil; graziellanas.silva@gmail.com (G.S.); jregrodrigues@gmail.com (J.R.); karollynams@hotmail.com (K.S.); rrsouza@ufs.br (R.S.)
2  Federal University of Bahia. R. Prof. Aristídes Novis, 2, Federação, Salvador 40210-630, BA, Brazil; keillascerqueira@hotmail.com
3  School of Chemical Engineering, Campinas State University, Cidade Universitária Zeferino Vaz. Av. Albert Einstein 500, Campinas 13083-852, SP, Brazil
*  Correspondence: diegofcoelho@gmail.com
†  These authors contributed equally to this work.

**Abstract:** Microalgae have a high growth rate, high $CO_2$ absorption capacity, and high content of chlorophyll, proteins, vitamins, mineral salts, carbohydrates, antioxidant substances, and fatty acids. In recent years, *Chlorella vulgaris* has been widely used as a feedstock for producing third-generation biofuels, such as bioethanol. Thus, this work aims to develop a strategy to increase the production scale of the microalgae *Chlorella vulgaris* grown in distilled reused water, supplemented with a modified BG-11 medium, to use biomass in the production of bioethanol. The total cultivation of 72 L presented a concentration of 0.415 $g \cdot L^{-1} \cdot d^{-1}$, with 61.32 g of final biomass. To improve carbohydrate extraction, the biomass was pre-treated with sulfuric acid at different concentrations (1.5% and 3% *v/v*). The hydrolyzed solution was supplemented with YPD (yeast extract peptone dextrose) medium and inoculated with *Saccharomyces cerevisae* yeast, initiating fermentation. In each sample, the Brix degree, cell concentration, reducing sugar concentration, and alcohol content were analyzed. The sample pre-treated with sulfuric acid 1.5% *v/v* was the one that presented the best result, with alcohol content after distillation of 68 °GL (Gay-Lussac). It appears that the cultivation of the microalgae *Chlorella vulgaris* in scale-up, with reused water, has high potential in the production of third-generation biofuel.

**Keywords:** open reactor; cultivation; *Chlorella vulgaris*; carbohydrates; bio-ethanol

## 1. Introduction

Microalgae are prokaryotic and eukaryotic photosynthetic microorganisms with a high growth rate in indoor or outdoor environments [1]. Microalgae have become very important, as they have accelerated metabolism and high efficiency for the photosynthetic conversion of sunlight into chemical energy. Under ideal growing conditions, they can achieve rapid growth and therefore shorten the biomass harvesting period. They produce various intracellular compounds such as proteins, carbohydrates, lipids, carotenoids, and polyunsaturated fatty acids [2]. Its intracellular compounds have antioxidant activity, in addition to being used for the production of renewable energy [3].

The microalgae culture medium requires micro- and macronutrients. Macronutrients (C, H, O, P, N, S, K, Mg, Si, and Fe) are responsible for the composition of the structures of biomolecules, contributing to the energy exchange process and the regulation of metabolic activities. Micronutrients (Mn, Cu, Zn, Mo, V, B, Ca, Na, Se, and Ni) are used as enzymatic cofactors in metabolic reactions; that is, they help and enable enzymatic catalysis [4]. The control of the concentration of nutrients, light, and temperature are factors that can

accelerate the growth of microalgae, and increase the accumulation of carbohydrates, an important component for the production of bioethanol, in addition to producing changes in the concentration of intracellular compounds [5].

There are several studies under development to optimize the use of renewable and sustainable resources to replace fossil fuels, such as the case of production in biogas industries using the plant digestate and design as part of a biorefinery for hybrid ethanol production, biogas, bio-oil, and fertilizers from biomass of *Chlorella vulgaris* [1]. In addition, it has been shown that *Chlorella* can grow in effluents of dairy, sewage sludge, and swine manure, and from this cultivation produce biofuels [4,6]. Another study using *Chlorella vulgaris* in biogas reinforces its characteristic in the treatment of residues and, as a consequence, the production of clean energy from biogas [3].

*Chlorella vulgaris* can also be used in corn silage, as cooking oil and mill residue for methane production [7]. Thus, the use of microalgae has gained space in the production of biofuels, since it has easy cultivation and adaptability to the most diverse environments such as saline, brackish, freshwater or even effluents [6,8].

Biofuels emerged to be a promising substitute for conventional fuels since they have lower greenhouse gas emissions. They are classified into 1st generation (using food crops), 2nd generation (using lignocellulosic biomass as raw material), and 3rd generation (using microalgae as raw material). The 3rd generation is more prominent, due to the 1st generation using food (sugarcane, soy derivatives, corn), thus requiring the use of several hectares of land, thus compromising agriculture. The 2nd generation requires pre-treatment, leading to an increase in production cost [9].

One of the main disadvantages of bioethanol production from lignocellulosic biomass is the excess energy spent in pre-treatment to break the barrier that lignin forms around cellulose and hemicellulose. Thus, it is necessary to break the cell wall to release the sugars. This rupture can be performed by acid hydrolysis, ultrasound, and autoclave. After the pre-treatment, the biological treatment also takes place, where the cellulose is decomposed into fermentative sugars. Thus, the cost of converting it into biofuel is very high [10,11]. Thus, microalgae can be recognized as an alternative for the production of biofuels, as they do not require arable land or drinking water and can be cultivated throughout the year [5]. In addition, microalgae have a $CO_2$ fixation capacity ten times more efficiently than plants and produce 30 to 100 times more energy per hectare when compared to crops [5] having the advantage of carbohydrates without lignin; that is, there is a higher content of fermentable sugars, allowing hydrolysis and rapid and effective release of sugar [12].

*Chlorella vulgaris* is an example of microalgae with high levels of carbohydrates and can accumulate up to 35–55 % of dry biomass [4,12,13]. This carbohydrate is the essential compound for bioethanol production; the greater the accumulation, the greater the production of bioethanol through fermentation. In addition, it has excellent stress adaptability of nutrients, light and temperature, favorable growth, and good nutrient removal in different types of effluents [14]. The bioethanol production process using microalgae generally involves nutritional stress (eg, nitrogen, nitrate, and phosphate deprivation) [15], cultivation in open reactors, and the reuse of effluents. The open reactor in the cultivation of microalgae has the advantages of being simple to build, affordable to acquire, requiring low maintenance, and ideal for cultivation in large proportions [8].

To occur fermentation and subsequent production of alcohol is necessary for the biomass to have a high content of carbohydrates. The accumulation of carbohydrates in microalgae occurs through the fixation of carbon dioxide as glucose and other sugars during photosynthesis, through the Calvin cycle [13].

The production of bioethanol occurs through the biological fermentation of sugar. To release the sugars from the microalgae biomass, it is necessary to break the cell walls, as the carbohydrates are concentrated mainly on the inner walls of the cell [5,8]. With this, a pre-treatment step is required to break through the cell walls and release its contents [4,16].

Treatments for cell wall rupture can be classified into mechanical (ball mill and ultrasound) and non-mechanical (osmotic shock, heating, drying, solvents, acids). The difference

between these treatments is that the mechanic uses equipment and the non-mechanical cell comes into contact with chemical substances [4,17].

The main carbohydrates from microalgae are starch and cellulose. Starch is composed of glucose considered propitious in the production of bioethanol, but hydrolysis is necessary since the carbohydrate is a complex compound needing to decompose into monosaccharide (glucose, mannose, xylose, galactose, fructose, and arabinose) for the fermentation process [12]. Fermentation occurs through anaerobic degradation of glucose by microorganisms, such as yeasts for bioethanol production, in various products obtaining energy in the form of ATP [12].

There are three stages involved in the production of bioethanol, including pre-treatment of biomass, hydrolysis of polysaccharides into simple sugars and conversion of these sugars into bioethanol through alcoholic fermentation [6,18].

Hydrolysis to be successful needs a pre-treatment, as it will break down polysaccharide molecules into fermentable sugars, then fermentation takes place, where there will be the transformation of sugars into bioethanol [18].

Fermentation occurs through the conversion of simple sugars into alcohol using a microorganism, usually using *Saccharomyces cerevisae*. Temperature and pH influence fermentation, where the best pH for fermentation using *Saccharomyces cerevisae* is between 4 and 5.5, and the ideal temperature of yeast growth is in the range of 25 to 35 °C [4]. Finally, the final phase is the distillation process, because the final mixture is composed of water and ethanol, making it necessary to apply the distillation process to separate the bioethanol [18].

Bioethanol produced from biomass can be used directly or mixed with gasoline in the engine (E-10 contains 10% bioethanol and 90% gasoline or E-85 contains 85% bioethanol and 15% gasoline). It has high octane content and combustion rate, so the evaporation temperature is higher than that of gasoline. It has a specific gravity of 0.79 $kg \cdot dm^{-3}$, the vapor pressure of 50 mmHg, the boiling temperature of 78.5 °C, and the molecular weight of 46.1 $g \cdot mol^{-1}$ [18].

This work aims to optimize the cultivation of the microalgae *Chlorella vulgaris* using a grown in distilled reused water, supplemented with modified BG-11 medium, to use the biomass in the production of bioethanol. The specific objectives of the present study are to (i) perform cultivation of microalgae in an open reactor applying the best condition previously seen on the shelf; (ii) produce bioethanol by submerged fermentation from the sugars obtained from hydrolysis; (iii) analyze alcohol content, reducing sugars, Brix and concentration; and (iv) compare the efficiency of sugar breakdown in alcohol formation.

## 2. Materials and Methods

### 2.1. Microalgae Chlorella Vulgaris: Cultivation and Culture Medium

The *Chlorella vulgaris* strain was obtained from the Laboratory of Environmental Biotechnology (LABAM) at the Federal University of Sergipe, Brazil. The strain was grown in a modified BG-11 medium [19] in a closed system in 2 L reactors on the shelf and monitored for 28 days in order to study the best growing conditions.

Following the results of the previous optimization study, when it was produced in a 2 L reactor, in an indoor glass photobioreactor with 2 L under constant luminous flux of 200 $\mu E \cdot m^{-2} s^{-1}$ (supplied by fluorescent lamps), constant forced aeration of 2 $L \cdot min^{-1}$ and a temperature of 26 ± 4 °C. The BG11 The BG-11 medium was prepared as described by [19]: 19 mg $Na_2CO_3$, 1500 mg $NaNO_3$, 5 mg $K_2HPO_4 \cdot 3H_2O$, 8 mg $MgSO_4 \cdot 7H_2O$, 22.65 mg $CaCl_2 \cdot 2H_2O$, 6 mg $C_6H_8O_7xFe_3 \cdot NH_3$, 0.736 mg $Na_2EDTA \cdot 2H_2O$, 6.4 mg $C_6H_8O_7$ and a trace metal solution of 3.0 mg $H_3BO_3$, 2.0 mg $MnCl_2 \cdot 4H_2O$, 0.20 mg $ZnSO_4 \cdot 7H_2O$, 0.4 mg $Na_2MoO_4 \cdot 2H_2O$, 0.13 mg $CuSO_4 \cdot 5H_2O$ and 0.066 mg $Co(NO3)_2 \cdot 6H_2O$. The best $NaNO_3$ concentration of BG-11 was 2300 $mg \cdot L^{-1}$ and the production time was 14 days; these conditions were replicated on a larger open scale. For this purpose, an aliquot of the strain (matrix) with a concentration of 1 $g \cdot L^{-1}$ was inoculated in open reactors with a volume of 18 L of reused water from the distiller's heat exchange system, with BG-11

medium; aeration was performed with atmospheric air continuously through a 4 L·min$^{-1}$ boyu SC3500 air pump (Chaozhou, China); and the solar lighting source with a photoperiod of 12 h of light: 12 h dark, with average daily solar irradiation of 4.500–5.102 kWh·m$^{-2}$, according to the Global Solar Atlas [20].

The stages of bioethanol production were the cultivation of microalgae, alcoholic fermentation, and simple distillation, as can be seen in Figure 1.

# Bioethanol production

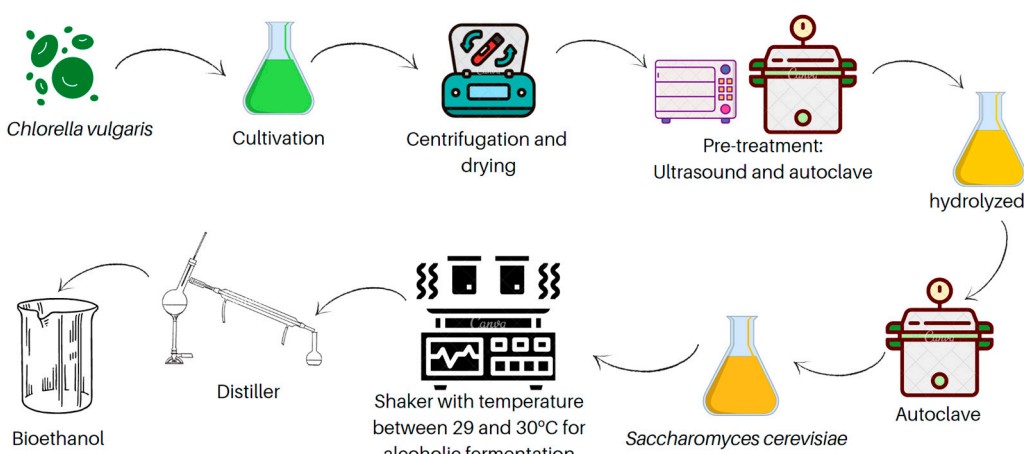

**Figure 1.** Stages of bioethanol production. Authors' own collection.

## 2.2. Alcoholic Fermentation

*Saccharomyces cerevisae* is generally used for alcoholic fermentation. The yeast was pre-inoculated in a YPD growth medium containing 20 g·L$^{-1}$ of glucose, 20 g·L$^{-1}$ of peptone, and 10 g·L$^{-1}$ of yeast extract [14]. After autoclaving the YPD medium, 1 g of yeast *Saccharomyces cerevisae* was added. Then, it was left in an incubation shaker (Certomat BS-1) at 200 rpm for 48 h at a temperature of 30 °C.

## 2.3. Carbohydrate Extraction (Pre-Treatment)

After 14 days, microalgae cultivation was stopped, and the supernatant was separated from the microalgae biomass using a centrifuge and then dried for 24 h at 80 °C. With the dry biomass, the extraction was performed by ultrasound and autoclave with different concentrations of solvent and 5 g of powdered microalgae was used together with 100 mL of sulfuric acid 1.5% or 3% (*v/v*) (methodology adapted from [21]), thus having two distinct experiments. The ultrasound treatment was performed for 1 h at a fixed temperature of 60 °C. The ultrasonic bath (USC 1400) was used at a frequency and power of 40 kHz and 135 W, respectively, a methodology adapted from [22]. The samples heated in an autoclave were kept at 120 °C for 30 min [23].

## 2.4. Bioethanol Characterization

The experiments were characterized as follows: distilled alcohol content according to the NBR 13920 standard [24] and determination of reducing sugars by the DNS method (3,5-dinitro salicylic acid) proposed by [25]. Brix content was measured using a refractometer (LH-T32, China). The concentration of cells in the fermentation medium was determined using a UV spectrophotometer (Kasuaki, São Paulo, Brazil), measuring the absorbance at a wavelength of 570 nm.

The ethanol obtained during fermentation (59 h) and the distillate were characterized using a 0.02 mL sample, which was diluted to 5.0 mL with distilled water, following the

procedure mentioned in the calibration curve. The determination of alcohol content in °GL was obtained by Equation (1).

$$E = \frac{5000 \times E'}{0.78934 \times 1000},$$ (1)

The variable E is the ethanol content in the sample in mL·100 mL$^{-1}$ (°GL), E′ is mg of ethanol in the aliquot (as read from the standard curve), 5000 is the aliquot conversion factor from 0.020 mL to 100 mL, 1000 is the conversion factor from mg to g, and 0.78934 is the density of ethanol at 20 °C, in g·mL$^{-1}$.

The construction of the standard curve was based on the standard ethanol solution, and with it, Equation (2) was obtained with $R^2$ = 0.957.

$$Y = 0.0014X + 0.0016$$ (2)

The variable X corresponds to the concentration in g $\times$ L$^{-1}$ of ethanol and Y to the absorbance value. With this equation, it was possible to calculate the bioethanol concentration.

The determination of reducing sugars was based on the reduction of 3-amino-5-nitrosalicylic acid in which the oxidation of the sugar's aldehyde group to a carboxylic acid group occurs. With that, the calibration curve was constructed where the following Equation (3) with $R^2$ = 0.983 was obtained.

$$Y = 0.7305X - 0.0674$$ (3)

The variable X corresponds to the concentration in g·L$^{-1}$ of total reducing sugars and Y to the absorbance value. Through this equation, it was possible to calculate the concentration of total reducing sugars in the analyzed samples.

The concentration of cells in the fermentative medium was determined with a spectrophotometer, measuring the absorbance at a wavelength of 570 nm, which was correlated with the values obtained through a previously constructed calibration curve, where the following was obtained from Equation (4) with $R^2$ = 0.932.

$$Y = 0.4398X + 0.4789$$ (4)

The variable X corresponds to the concentration in g·L$^{-1}$ and Y to the absorbance value. Using this equation, it was possible to calculate the yeast cell concentration.

### 2.5. Simple Distillation for Obtaining Bioethanol

The supernatant of the fermented medium was transferred to a 250 mL flask, which was coupled to the simple distillation system using water as the solvent. At the beginning of the process, the first drops of distillate were discarded the distillation continued for approximately 3 h.

## 3. Results

### 3.1. Biomass Yield

The cultivation in 72 L was considered to have similar characteristics to the culture medium in a closed environment. The collections were carried out on alternate days to compare the productivity and the maximum concentration obtained. Thus, the productivity in open culture in the batch of 4 bioreactors of 18 L with a total of 72 L was on average 0.415 g·L$^{-1}$·d$^{-1}$ and a maximum concentration of 30 g·L$^{-1}$, and has a total dry biomass of 61.32 g, and in the individual reactor (18 L) a total of 15.33 g·

In this test, 1 bioreactor of 18 L was used to monitor the growth kinetics, as can be seen in Figure 2, although to optimize the production of alcohol 72 L of microalgae culture was grown.

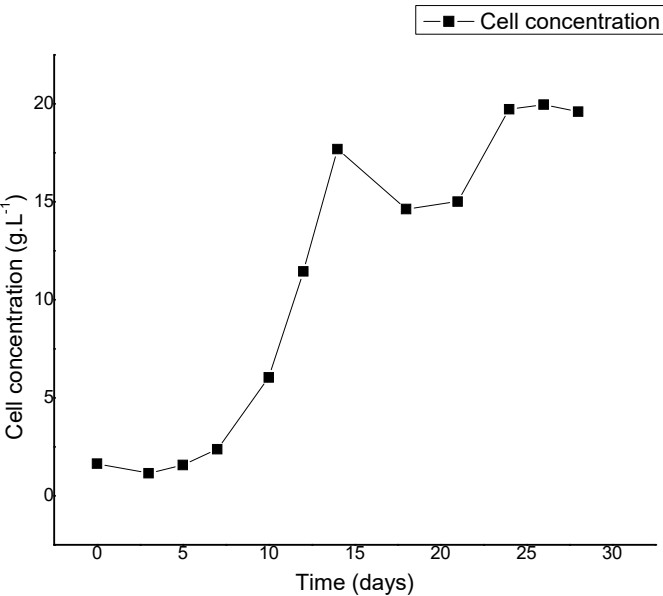

**Figure 2.** Growth kinetics curve over the 28 days of cultivation, whose growth peak occurred at 14 days.

At the end of the cultivation of the 4 individual bioreactors, the total biomass referring to the 72 L was decanted, centrifuged, and dried. Then, for the extraction of sugars were used 2 types of sulfuric acid concentrations were in order to evaluate the best condition presented in the following topics.

*3.2. Pre-Treatment Evaluation*

The two experiments were analyzed for Brix and the concentration of reducing sugars. The samples showed Brix equal to 3 and reducing sugars with an average of 0.6 g·L$^{-1}$, which can be seen in Table 1.

**Table 1.** Result of samples without and with pre-treatment.

| Sample | Concentration Sulfuric Acid (%) | °Brix | | ART (g·L$^{-1}$) | |
|---|---|---|---|---|---|
| | | Before | After | Before | After |
| 1 | 1.5 | 3 | 15.5 | 0.69 | 1.80 |
| 2 | 3 | 3 | 9 | 0.51 | 1.01 |

*3.3. Fermentation of the Hydrolyzate*

The two experiments, after going through the pre-treatment, were placed in the shaker, and analyses of the concentration of reducing sugars, alcohol content, Brix degree, and cell concentration were carried out periodically.

Using the calibration curve, it was possible to obtain the equation of the straight line to discover the concentration in g·L$^{-1}$; with this, using Equations (2)–(4) of the calibration curve, it was possible to obtain the concentration values of ethyl alcohol and of total reducing sugars and cell concentration, respectively, during the entire hydrolyzate fermentation process, which lasted 59 h.

With this, a graph was drawn for each behavioral condition of time about the concentration in g·L$^{-1}$ of alcohol, total reducing sugars, and cell concentration. These graphs can be seen in Figures 3 and 4.

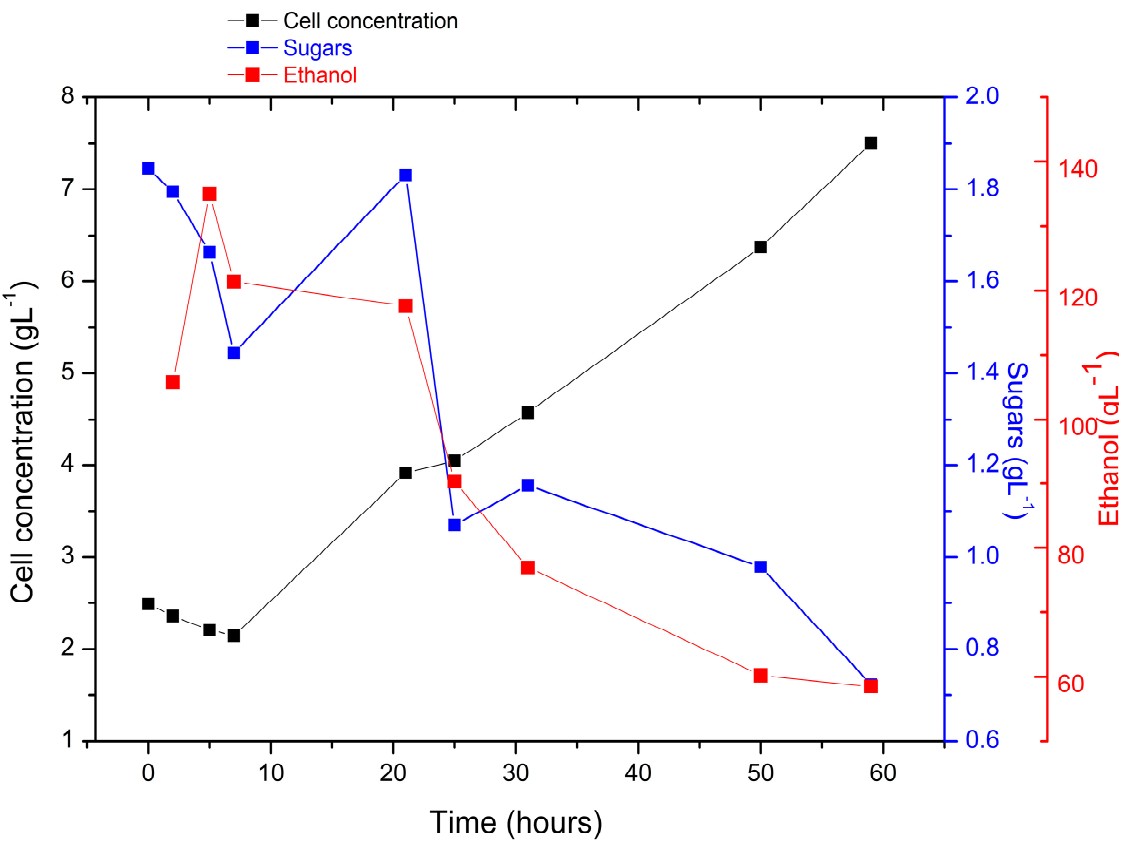

**Figure 3.** Alcohol concentration, reducing sugars, and cellular concentration over time in the sulfuric acid concentration of 1.5% (*v/v*) (sample 1).

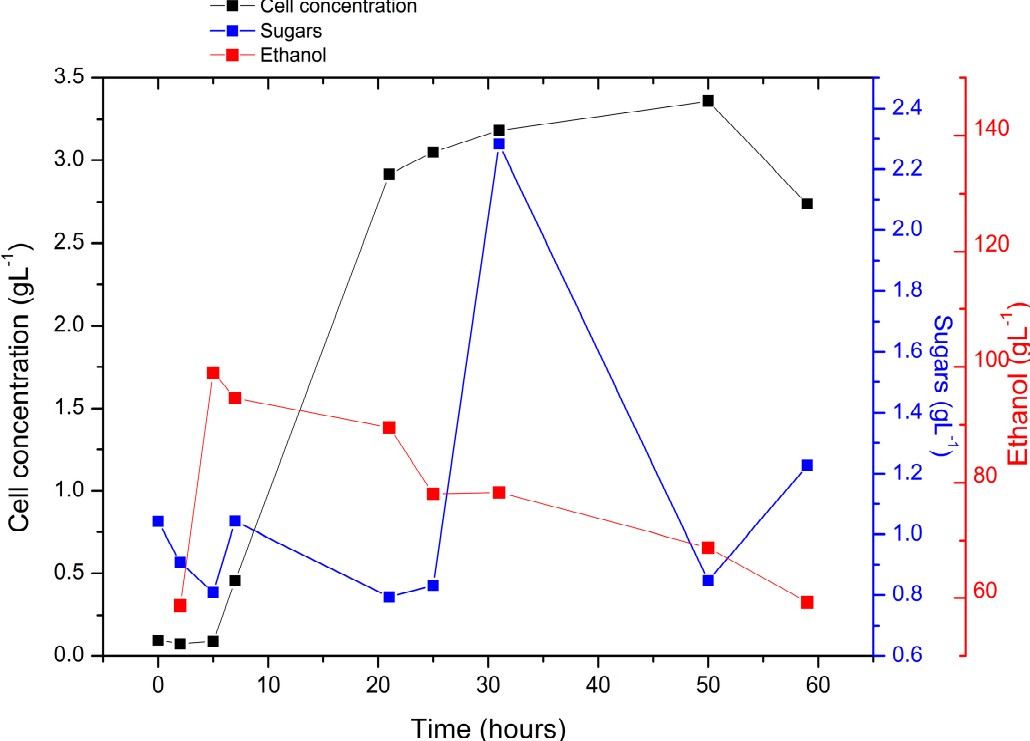

**Figure 4.** Alcohol concentration, reducing sugars, and cellular concentration over time in the sulfuric acid concentration of 3% (*v/v*) (sample 2).

In Figures 5–7, cell concentration, alcohol concentration, and sugar concentration can be compared for samples 1 and 2 with standard deviations, respectively.

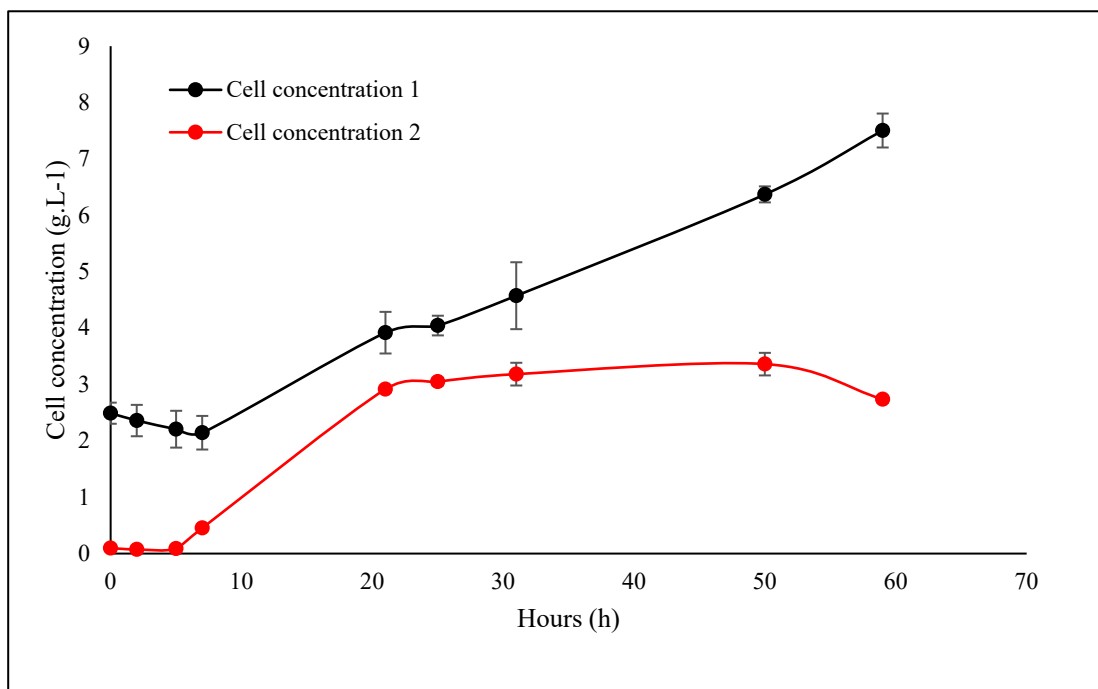

**Figure 5.** Cell concentration over time in the sulfuric acid of 1,5 % (*v/v*) sample 1 and 3 % (*v/v*) sample 2.

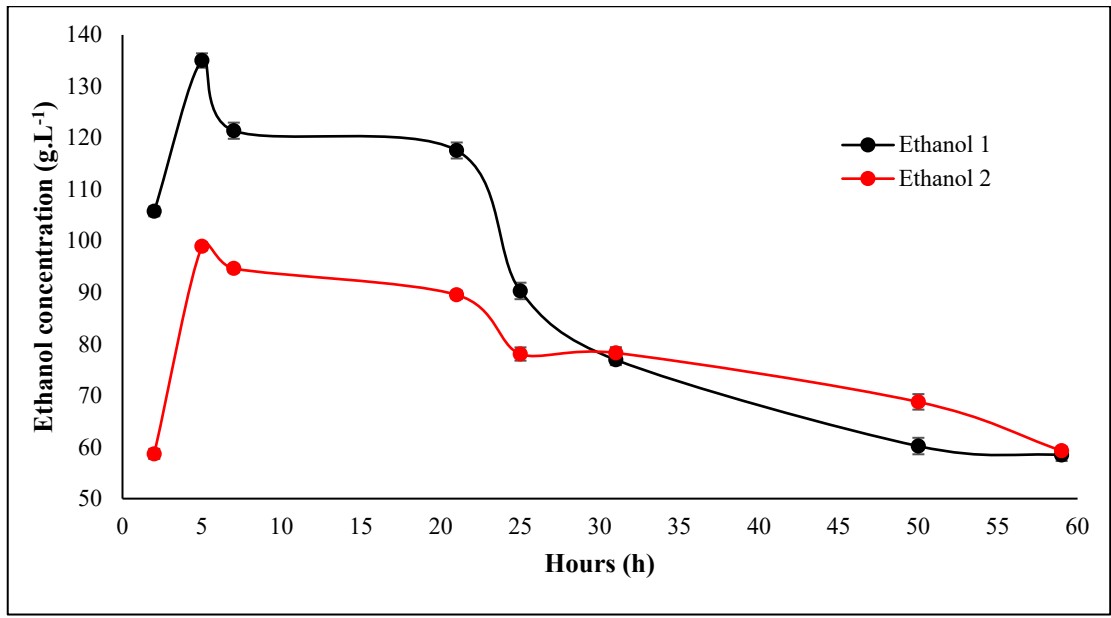

**Figure 6.** Ethanol concentration over time in the sulfuric acid of 1,5 % (*v/v*) sample 1 and 3 % (*v/v*) sample 2.

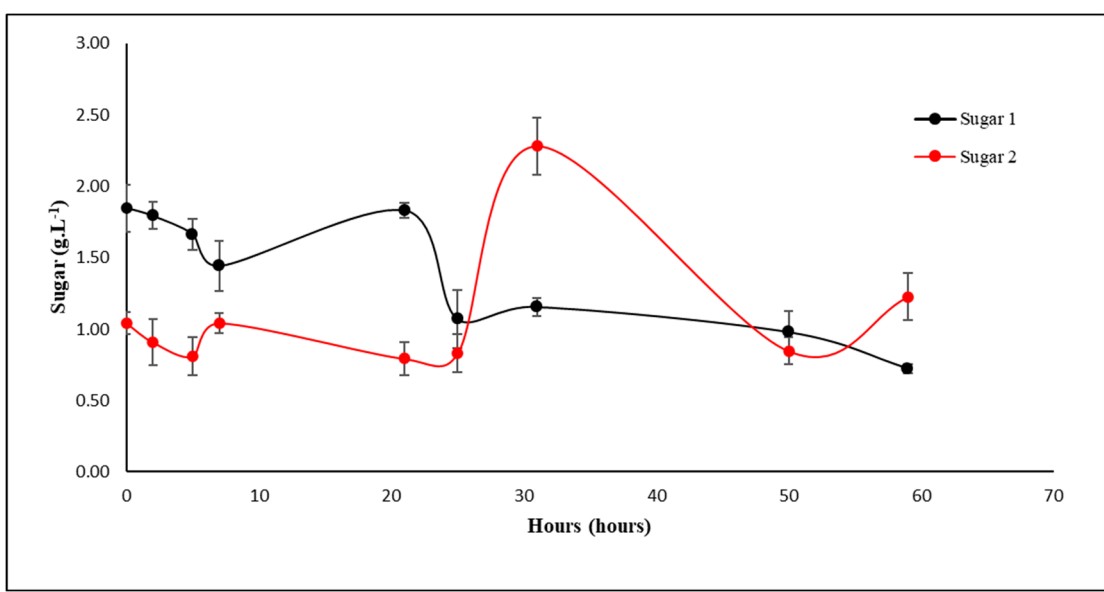

**Figure 7.** Sugar concentration over time in the sulfuric acid of 1,5 % (*v/v*) sample 1 and 3 % (*v/v*) sample 2.

### 3.4. Alcoholic Content

In Table 2 it is possible to compare the maximum concentrations, their times, and also the ethanol content in the sample in mL; 100 mL$^{-1}$ (°GL) using Equation (1) to perform the calculation.

**Table 2.** Summary of alcohol content results.

| Sample | Alcohol Concentration (g·L$^{-1}$) | Time (h) | Ethanol Content (°GL) |
|---|---|---|---|
| 1 | 136.60 | 5 | 68 |
| 2 | 98.17 | 5 | 49 |

## 4. Discussion

### 4.1. Biomass Yield

The result of microalgae cultivation in an open reactor was satisfactory, as it was possible to obtain large amounts of biomass about the reactor grown on a shelf. In addition, the open system cultivation operation is simple and inexpensive. Thus, the productivity of microalgae was satisfactory using 4 reactors of 18L and using hot water from the distiller.

Sakarika and Kornaros [26] reported a productivity of *Chlorella vulgaris* of 0.828 g·L$^{-1}$·d$^{-1}$, with batch cultivation. The referred study already presented productivity of 0.415 g·L$^{-1}$·d$^{-1}$ in outdoor cultivation with distilled reused water.

### 4.2. Pre-Treatment Evaluation

As can be seen, the pre-treatment with sulfuric acid (1.5 % and 3 % *v/v*) in conjunction with ultrasound and autoclave increased Brix and the concentration of total reducing sugars. Thus, it can be observed in Table 1 that the concentration of sulfuric acid at 1.5 % in the pre-treatment showed better results. This statement is also valid when compared with the work of Ngamsirisomsakul [27], which uses concentrations of 0, 0.75, and 1.5 % (*v/v*) of sulfuric acid, and the concentration that showed the best result in the pre-treatment was 1.5 % (*v/v*).

*4.3. Fermentation of the Hydrolysate*

From the graph in Figure 6, it can be seen that the highest alcohol concentration was 136.60 g·L$^{-1}$ on sample 1, with fermentation time equal to 5 h, after which decay occurs. Therefore, there is no need to continue the experiment for long hours.

Over time there is a decay of sugar, proving that there is a release of sugar after extraction with pre-treatment, and over time this sugar is used by the yeast, thus producing ethyl alcohol, as can be seen in Figure 7.

A decay in yeast growth during the initial 7 h can also be seen in Figure 3, but then it starts to grow again, which is explained by the adaptation of the yeast to the medium, which is why the analysis continued and its behavior was verified. At 20 h, the yeast consumed more sugar, not to produce alcohol, but to maintain its growth over the 59 h of analysis.

In Figure 3, it is observed that sample 2 obtained an ethyl alcohol concentration of 98.17 g·L$^{-1}$ with a time equal to 5 h, as well as sample 1, without the need to spend a lot of time fermenting.

Comparing the graphs in Figures 3 and 4, it can be seen that there was a greater extraction of carbohydrates in sample 1 using the 1.5% sulfuric acid solution than in sample 2, which uses 3% sulfuric acid. However in this graph in Figure 4, one can also see an oscillation over time.

Figure 5 shows that unlike sample 1, sample 2 grows slowly in the first 7 h, but over time it grows very fast, and between 50 and 59 h, the cells begin to die. Still comparing samples 1 and 2, in the first, the growth is more than double in the second. It can be stated that the medium where the breakdown with 1.5% sulfuric acid was used was more conducive to growth than the medium with 3% sulfuric acid. That is a fair statement that the highest percentage of sulfuric acid negatively influences the metabolism of yeasts.

According to Acebu et al. [14] in their work with the production of bioethanol from *Chlorella vulgaris* grown in swine wastewater, the production realized a pre-treatment with 4% sulfuric acid and autoclave, which contributed to a maximum concentration of bioethanol of 4.2 g·L$^{-1}$. The result of the work with pre-treatment at 1.5% sulfuric acid followed by ultrasound and autoclave obtained a maximum concentration of sample 1 of 136.6 g·L$^{-1}$. Such a result may be due to the use of two pre-treatments or even due to the time spent in the autoclave.

*4.4. Alcohol Content*

According to Table 2, it was possible to obtain a Gay-Lussac degree of 68 in sample 1; that is, in the sample for every 100 mL, there are 68 mL of alcohol and 32 mL of water. Then, we took sample 2 with 49 °GL. However, both took the same time to reach their respective degrees. Therefore, to replicate on larger scales, sample 1 is the best choice, as it is the sample that obtained sulfuric acid with a concentration of 1.5% (*v*/*v*) as pre-treatment.

The work by Ngamsirisomsakul [27] with pre-treatment of *Chlorella* biomass with sulfuric acid at a concentration of 1.5% (*v*/*v*) showed a maximum concentration of bioethanol produced of 5.62 g·L$^{-1}$. While on the work under study, it obtained a maximum concentration of 136.60 g·L$^{-1}$. With this, it is possible to state that the research produced very satisfactory results.

In Brazil, ANP resolution N° 734 of 28 June 2018, chapter 2, art. 2, item VI defines that biofuels are any renewable biomass substance, which can be used in internal combustion engines or for another type of energy generation. According to Sakarika and Kornaros [28], the biomass of *Chlorella vulgaris* meets the requirements for most regions. As a result, bioethanol produced from *Chlorella vulgaris* biomass could be used in the future as a new source of energy.

## 5. Conclusions

The microalgae productivity in an open reactor with a batch of 4 reactors adding up to 72 L was 0.415 g·L$^{-1}$·d$^{-1}$ and a maximum concentration of 30 g·L$^{-1}$, with a dry biomass

of 15.33 g in the separate reactor. When compared to the 2 L reactor on the shelf, which has a maximum concentration of 19.598 g·L$^{-1}$, a productivity of 0.642 g·L$^{-1}$·d$^{-1}$, and a dry biomass of 1.858 g, it is better to cultivate in an open reactor, in addition to being able to use water from the distiller with no need for autoclaving.

When comparing the two experiments, it can be concluded that the sample pre-treated with sulfuric acid 1.5 % *v/v* showed a greater potential for bioethanol production, with greater yeast growth and, consequently, greater production of ethanol with an ethyl alcohol content of 136.60 g·L$^{-1}$ (68 °GL) and fermentation time of 5h. It can be seen that the microalgae *Chlorella vulgaris*, cultivated on a larger scale with reused water, is a raw material with high potential in the production of third-generation biofuel.

**Author Contributions:** Conceptualization: G.S., K.C. and J.R.; investigation and conceived the research: G.S., K.C., J.R., K.S., R.S. and D.C.; writing—original draft preparation: G.S., K.C. and J.R.; writing—review and editing: G.S., K.C. and J.R. All authors have read and agreed to the published version of the manuscript.

**Funding:** This work was financially supported by CNPq (National Council for Scientific and Technological Development—Brazil) process identified by the number MCT–CNPq 304025/2010-0.

**Institutional Review Board Statement:** Not applicable.

**Informed Consent Statement:** Not applicable.

**Data Availability Statement:** Not applicable.

**Acknowledgments:** The authors gratefully acknowledge the support provided by our colleagues from the Laboratory of Environmental Biotechnology (LABAM). Special thanks to Roberto Rodrigues de Souza, the laboratory manager.

**Conflicts of Interest:** The authors declare no conflict of interest.

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
