# Peer review of "Cultivation of Microalgae Chlorella vulgaris in Open Reactor for Bioethanol Production"

_phycology, doi:10.3390/phycology3020021_

Round 1
Reviewer 1 Report
The paper titled “Cultivation Of Microalgae Chlorella vulgaris In Open Reactor For Bioethanol Production” deals with a developped strategy aimed to enhance the growth of Chlorella vulgaris at industrial scale by using reused water from a distillery to increase the production of biomass and the extraction of bioethanol from it. Overall the paper presents a solution of interest for the specific case investigated, which can potentially exploited at industrial scale. However, the manuscript should be refined to better focus the scope (which is not declared at all in the Introduction) and to amplify the discussion that in some points is extremely concise. Authors should in particular provide statistical analysis of data reported (totally missing) and in general address all the points that are reported below.
In the title OpenReactor are two separated words Open Reactor
In the abstract, line 18 grown in distilled reused water
In introduction line 73, excellent adaptability to what? You should clarify to which extreme condition this microalga can be adapted (stress of nutrients? Light? Salinity?)
At the end of the Introduction, based on the state of the art mentioned above, you should clearly declarate what is the aim of this work and what you are going to show in the manuscript
In Material and Mathods (M&M) line 85, specify where University of Sergipe is located, that should be Brazil
In M&M line 86-87, specify in detail the characteristics of 2 L reactors, are you working with 2 L? or the working volume was lower than 2 L? What is the material of the 2 L reactor? Glass? Plastic? How the 2 L reactors were illumated? Where they blubbled? With air? Air and CO2 mixed? Provide all these details or if they have been provided in your previous pubblication add reference leading the reader to it
In M&M starting from line 89 and in the whole manuscript so on the units of measure are reported as mg.L-1, 4 L.min-1, kW.h.m-2 and so on. I have never seen this way to report the units? It is correct? Is the point after the unit (mg, L, kW) required? I don’t believe so
In M&M at the end of the paragraph 2.1 is reported the Figure 1. Figure showing the experimental results have to be reported in Results section (3) not in M&M. In addition, as explained in lines 96-97, this figure shows the cell concentration of C. vulgaris growth after 28 days obtained by 4 reactors of 18 L each for a total of 72 L. Is this growth curve obtained as an average of the 4 reactors? If yes where are the error bars in the black squares reporting the standard deviation of the 4 replicates? Same for figure 2 and 3
At the end of M&M after the paragraph 2.5 I think that it should be added another paragraph 2.6 for Statistical analysis explaining how many replicates you have done of your experiments (3, 4?) and how have been processed the standard deviations that are missing in Figure 1, 2 and 3
In paragraph 3.4 line 188, what is the meaning of mL.100mL-1
In discussion line 199, as it can be
In discussion, lines 227-229 and 231-232 the same words are used in the same sentence (it can be stated and production of bioethanol). Rephrase it avoiding repetition
In abstract (line 29), line 128, line 188 the letters GL are used without to specify their meaning that is Gay Lussac which appear only in line 239 for the first time. You should explain the full meaning the first time it is used (in the abstract) and later you can report it as GL throughout the whole manuscript
In line 247 change obtained in produced or showed
In the reference list the name of all microorganisms (algae and bacteria) have to be reported in Italics with the second name in lowercase. So please check it for references # 1, 2, 5, 9, 14, 15, 18, 20, and 23. The same have to be done for the whole manuscript (i.e. line 84, 232, 244, and 260)
In the reference list the name of the journal have to be reported in full not abbreviated (check ref # 22) and when in full with uppercase of the first letter in the words (check # 12 and 20)
Overall the English is acceptable but in some parts (whcih have been marked in the manuscript) authors should refine or rephrase the whole sentence because there are repetitions that make hard the comprehension of the meaning.
Author Response
Response to Reviewer 1 Comments
In this manuscript, the authors investigate the process of bioethanol production from the microalgae Chlorella vulgaris in open reactors using distiller water. The main objective was the production of bioethanol to optimize the cultivation of the microalgae Chlorella vulgaris using culture in distilled water of reuse, supplemented with modified BG-11 medium, to use the biomass in the production of bioethanol. The authors also determined as specific objectives of the present study, to carry out the cultivation of microalgae in an open reactor applying the best condition already seen on the shelf; to produce bioethanol by submerged fermentation from the sugars obtained by hydrolysis; analyze alcohol content, reducing sugars, Brix and concentration; and compare the efficiency of the sugar break in the formation of alcohol.
First, we would like to thank you for your valuable comments and suggestions, which helped us to improve the manuscript. Below we try to address all the points, which you have indicated in your revision. The new/changed text is tracked in the manuscript.
Point 1: In the title OpenReactor are two separated words Open Reactor
Response 1: The article has been revised and changes were made to make the manuscript clearer. Regarding the repeated information, adjustments were made in lines 2-3.
Point 2: In the abstract, line 18 grown in distilled reused water
Response 2: Adjustments were made in line 18.
Point 3: In introduction line 73, excellent adaptability to what? You should clarify to which extreme condition this microalga can be adapted (stress of nutrients? Light? Salinity?)
Response 3: Chlorella vulgaris is an example of microalgae with high levels of carbohydrates, in addition to excellent stress adaptability of nutrients, light, and temperature. Adjustments were made in lines 84 - 89.
Point 4: At the end of the Introduction, based on the state of the art mentioned above, you should clearly declarate what is the aim of this work and what you are going to show in the manuscript.
Response 4: This work aims to optimize the cultivation of the microalgae Chlorella vulgaris using a grown in distilled reused water, supplemented with modified BG-11 medium, to use the biomass in the production of bioethanol. The specific objectives of the present study are to: (i) Perform cultivation of microalgae in open reactor applying the best condition previously seen on shelf; (ii) Produce bioethanol by submerged fermentation from the sugars obtained from hydrolysis; (iii) Analyse alcohol content, reducing sugars, Brix and concentration; (iv) Compare the efficiency of sugar breakdown in alcohol formation. Adjustments were made in lines 137 - 143.
Point 5: In Material and Mathods (M&M) line 85, specify where University of Sergipe is located, that should be Brazil.
Response 5: The Chlorella vulgaris strain was obtained from the Laboratory of Environmental Biotechnology - LABAM at the Federal University of Sergipe - Brazil. Adjustments were made in line 147.
Point 6: In M&M line 86-87, specify in detail the characteristics of 2 L reactors, are you working with 2 L? or the working volume was lower than 2 L? What is the material of the 2 L reactor? Glass? Plastic? How the 2 L reactors were illumated? Where they blubbled? With air? Air and CO2 mixed? Provide all these details or if they have been provided in your previous pubblication add reference leading the reader to it.
Response 6: Following the results of the previous optimization study, when it was produced in a 2L reactor, in an indoor glass photobioreactor with 2 L under constant luminous flux of 200 μE.m-2.s-1 (supplied by fluorescent lamps), constant forced aeration of 2 L.min-1 and temperature of 26 ± 4ºC. The BG11 The BG-11 medium was prepared as described by: 19 mg Na2CO3, 1500 mg NaNO3, 5 mg K2HPO4·3H2O, 8 mg MgSO4 ·7H2O, 22.65 mg CaCl2· 2H2O, 6 mg C6H8O7xFe3·NH3, 0.736 mg Na2EDTA.2H2O, 6.4 mg C6H8O7 and a trace metal solution of 3.0 mg H3BO3, 2.0 mg MnCl2·4H2O, 0.20 mg ZnSO4·7H2O, 0.4 mg Na2MoO4. 2H2O, 0.13 mg CuSO4·5H2O and 0.066 mg Co(NO3)2.6H2O. The best NaNO3 concentration of BG-11 was 2300 mg.L-1, and the production time was 14 days, these conditions were replicated in a larger open scale. Adjustments were made in lines 150 - 159.
Point 7: In M&M starting from line 89 and in the whole manuscript so on the units of measure are reported as mg.L-1, 4 L.min-1, kW.h.m-2 and so on. I have never seen this way to report the units? It is correct? Is the point after the unit (mg, L, kW) required? I don’t believe so.
Response 7: All the measures were adjusted along the article.
Point 8: In M&M at the end of the paragraph 2.1 is reported the Figure 1. Figure showing the experimental results have to be reported in Results section (3) not in M&M. In addition, as explained in lines 96-97, this figure shows the cell concentration of C. vulgaris growth after 28 days obtained by 4 reactors of 18 L each for a total of 72 L. Is this growth curve obtained as an average of the 4 reactors? If yes where are the error bars in the black squares reporting the standard deviation of the 4 replicates? Same for figure 2 and 3.
Response 8: No, the calibration curve was built with the strain of a single bioreactor of 18L and was used to monitor the growth kinetics, as can be seen in Figure 1, although to optimize the production of alcohol, it was used 72L of microalgae grown.
Figure 2 was modified to the results section.
Regarding the data of Figures 3 and 4, at the end of the cultivation of the 4 individual bioreactors, the total biomass referring to the 72L was decanted, centrifuged, and dried. Then, for the extraction of sugars were used 2 types of sulfuric acid concentrations were in order to evaluate the best condition presented in Figures 3 and 4.
Adjustments were made in lines 232 - 234.
Point 9: At the end of M&M after the paragraph 2.5 I think that it should be added another paragraph 2.6 for Statistical analysis explaining how many replicates you have done of your experiments (3, 4?) and how have been processed the standard deviations that are missing in Figure 1, 2 and 3.
Response 9: The analysis referring to figures 1, 2, and 3 was performed with only a single replication, in view of being a base study for bioethanol production from microalgae cultivation. We plan to carry out future trials with replicas for each analysis.
Point 10: In paragraph 3.4 line 188, what is the meaning of mL.100mL-1
Response 10: It is the unit of ºGay Lussac that represents 1 mL of absolute alcohol contained in 100 mL of the hydroalcoholic mixture.
Point 11: In discussion line 199, as it can be
Response 11: Adjustments were made in line 312.
Point 12: In discussion, lines 227-229 and 231-232 the same words are used in the same sentence (it can be stated and production of bioethanol). Rephrase it avoiding repetition.
Response 12: Adjustments were made in line 312 and 314 - 317.
Point 13: In abstract (line 29), line 128, line 188 the letters GL are used without to specify their meaning that is Gay Lussac which appear only in line 239 for the first time. You should explain the full meaning the first time it is used (in the abstract) and later you can report it as GL throughout the whole manuscript
Response 13: Adjustments were made in line 26.
Point 14: In line 247 change obtained in produced or showed
Response 14: Adjustments were made in line 330-331.
Point 15: In the reference list the name of all microorganisms (algae and bacteria) have to be reported in Italics with the second name in lowercase. So please check it for references # 1, 2, 5, 9, 14, 15, 18, 20, and 23. The same have to be done for the whole manuscript (i.e. line 84, 232, 244, and 260)
Response 15: Adjustments were made.
Point 16: In the reference list the name of the journal have to be reported in full not abbreviated (check ref # 22) and when in full with uppercase of the first letter in the words (check # 12 and 20).
Response 16: Adjustments were made.
Reviewer 2 Report
The Manuscript ID: phycology-2370062 “CULTIVATION OF MICROALGAE Chlorella vulgaris IN OPEN REACTOR FOR BIOETHANOL PRODUCTION” requires revision before accepted for publication. The specific comments are given below.
1. In the abstract, present the most important numerical results.
2. Provide significant words which are more relevant to the work in a logical sequence as ‘keywords’.
3. The "Introduction" section should follow the state of the art of this field and review what has been done, for supporting the research gap and the significance of this study. Please improve the state of the art overview, to clearly show the progress beyond the state of the art.
4. In the introduction, it is worth mentioning what other biofuels were made from Chlorella vulgaris eg. https://doi.org/10.3390/en15082912
https://doi.org/10.1007/s10811-016-0796-5
https://doi.org/10.1016/j.biortech.2017.05.194
https://doi.org/10.1007/s12649-016-9667-1
https://doi.org/10.1016/j.enconman.2014.11.050
https://doi.org/10.1002/ep.12294
https://doi.org/10.1016/j.biortech.2018.11.017
https://doi.org/10.1016/j.biortech.2017.05.194
https://doi.org/10.1016/j.biortech.2020.122793
5. In the last paragraph of the introduction, clearly indicate the aim of research and hypothesis.
6. Correct the multiplication sign, e.g. ln 89, 91, 94, 95...
7. Were the studies repeated? Add standard deviations to the results of the presented research.
8. Please indicate the manufacturer, city, country when mentioning the equipment.
9. Statistical research is very important in experiments. How were the significances of the differences between the variables determined? Complete the methodology.
10. Expand the discussion significantly and compare the results of your research with those obtained by other authors.
11. It is also recommended to discuss and explain what should be the appropriate policies based on the findings of this study.
12. Adapt the bibliography to the requirements of the MDPI.
Author Response
Response to Reviewer 2 Comments
In this manuscript, the authors investigate the process of bioethanol production from the microalgae Chlorella vulgaris in open reactors using distiller water. The main objective was the production of bioethanol to optimize the cultivation of the microalgae Chlorella vulgaris using culture in distilled water of reuse, supplemented with modified BG-11 medium, to use the biomass in the production of bioethanol. The authors also determined as specific objectives of the present study, to carry out the cultivation of microalgae in an open reactor applying the best condition already seen on the shelf; to produce bioethanol by submerged fermentation from the sugars obtained by hydrolysis; analyze alcohol content, reducing sugars, Brix and concentration; and compare the efficiency of the sugar break in the formation of alcohol.
First, we would like to thank you for your valuable comments and suggestions, which helped us to improve the manuscript. Below we try to address all the points, which you have indicated in your revision. The new/changed text is tracked in the manuscript.
Point 1: In the abstract, present the most important numerical results.
Response 1: Microalgae have a high growth rate, high CO2 absorption capacity, and high content of chlorophyll, proteins, vitamins, mineral salts, carbohydrates, antioxidant substances, and fatty acids. In recent years, Chlorella vulgaris has been widely used as a feedstock for producing third-generation biofuels, how bioethanol. Thus, this work aims to develop a strategy to increase the production scale of the microalgae Chlorella vulgaris grown in distilled reused water, supplemented with a modified BG-11 medium, to use biomass in the production of bioethanol. The total cultivation of 72L presented a concentration of 0.415 g.L-1d-1, with 61.32g of final biomass. To improve carbohydrate extraction, the biomass was pre-treated with sulfuric acid at different concentrations (1.5% and 3% v/v). The hydrolyzed solution was supplemented with YPD (Yeast extract-Peptone-Dextrose) medium and inoculated with Saccharomyces cerevisae yeast, initiating fermentation. In each sample, the Brix degree, cell concentration, reducing sugar concentration, and alcohol content were analyzed. The sample pre-treated with sulfuric acid 1.5% v/v was the one that presented the best result, with alcohol content, after distillation, of 68 oGL (Gay Lussac). It appears that the cultivation of the microalgae Chlorella vulgaris, in scale-up, with reused water, has high potential in the production of third-generation biofuel.
Adjustments were made in lines 15, 19-20.
Point 2: Provide significant words which are more relevant to the work in a logical sequence as ‘keywords’.
Response 2: Open reactor; cultivation; Chlorella vulgaris, carbohydrates; bioethanol. Adjustments were made in line 29-30.
Point 3: The "Introduction" section should follow the state of the art of this field and review what has been done, for supporting the research gap and the significance of this study. Please improve the state of the art overview, to clearly show the progress beyond the state of the art.
Response 3: Adjustments were made throughout the "introduction" session.
Point 4: In the introduction, it is worth mentioning what other biofuels were made from Chlorella vulgaris eg. https://doi.org/10.3390/en15082912
https://doi.org/10.1007/s10811-016-0796-5
https://doi.org/10.1016/j.biortech.2017.05.194
https://doi.org/10.1007/s12649-016-9667-1
https://doi.org/10.1016/j.enconman.2014.11.050
https://doi.org/10.1002/ep.12294
https://doi.org/10.1016/j.biortech.2018.11.017
https://doi.org/10.1016/j.biortech.2017.05.194
https://doi.org/10.1016/j.biortech.2020.122793
Response 4: Adjustments were made throughout the "introduction" session.
Point 5: In the last paragraph of the introduction, clearly indicate the aim of research and hypothesis.
Response 5: This work aims to optimize the cultivation of the microalgae Chlorella vulgaris using a grown in distilled reused water, supplemented with modified BG-11 medium, to use the biomass in the production of bioethanol. The specific objectives of the present study are to: (i) Perform cultivation of microalgae in open reactor applying the best condition previously seen on shelf; (ii) Produce bioethanol by submerged fermentation from the sugars obtained from hydrolysis; (iii) Analyse alcohol content, reducing sugars, Brix and concentration; (iv) Compare the efficiency of sugar breakdown in alcohol formation. Adjustments were made in lines 137 - 143.
Point 6: Correct the multiplication sign, e.g. ln 89, 91, 94, 95....
Response 6: Adjustments were made throughout the article.
Point 7: Were the studies repeated? Add standard deviations to the results of the presented research.
Response 7: The analysis referring to figures 2, 3, and 4 was performed with only a single replication, in view of being a base study for bioethanol production from microalgae cultivation. We plan to carry out future trials with replicas for each analysis.
Point 8: Please indicate the manufacturer, city, country when mentioning the equipment.
Response 8: Adjustments were made throughout the article.
Point 9: Statistical research is very important in experiments. How were the significances of the differences between the variables determined? Complete the methodology.
Response 9: The analysis was performed with only a single replication, in view of being a base study for bioethanol production from microalgae cultivation. We plan to carry out future trials with replicas for each analysis.
Point 10: Expand the discussion significantly and compare the results of your research with those obtained by other authors.
Response 10: Adjustments were made throughout the article.
Point 11: It is also recommended to discuss and explain what should be the appropriate policies based on the findings of this study.
Response 11: Adjustments were made in line 332-336.
Point 12: Adapt the bibliography to the requirements of the MDPI.
Response 12: Adjustments were made.
Reviewer 3 Report
Critical remarks and suggestions for the manuscript ID phycology-2370062:
The abstract needs a major overhaul. In the beginning there are too many generalities that do not relate to the content of the manuscript. Too much space is devoted to the methodological assumptions and little to the results obtained, the most important discoveries and achievements. These proportions must be changed.
Add more keywords that characterize the content of the manuscript.
The introduction section should precisely present what the research brings to the commonly known knowledge, what was the inspiration for the works, what research hypotheses they wanted to verify, what is the progress of these studies in relation to the ones carried out so far.
There is no clearly and specifically formulated research objective, which should be clear and distinct at the end of the Introduction chapter.
Please enter the Experiment design or Research organization subchapter in Methodology, where exactly, step by step, the division and sequence of experimental works will be presented. Please consider the block diagram.
Chapters: Fermentation, Characterisation, Distillation should be corrected with precise names, because it is not entirely clear what the mentioned processes are about.
Please describe what methods and tests of statistical data analysis were used. A well-conducted statistical analysis of the results is a necessary condition for the credibility of the presented research and formulated conclusions.
Figure 1. Growth kinetics curve over the 28 days of cultivation, whose growth peak occurred 99 at 14 days and the description should be moved to section 3. Results
The results should be presented not only in terms of average values, but also with standard deviations. This should be completed in charts, tables and content.
Discussion of the results and discussion is very poor and laconic, it requires deep supplementation and development.
Author Response
Response to Reviewer 3 Comments
In this manuscript, the authors investigate the process of bioethanol production from the microalgae Chlorella vulgaris in open reactors using distiller water. The main objective was the production of bioethanol to optimize the cultivation of the microalgae Chlorella vulgaris using culture in distilled water of reuse, supplemented with modified BG-11 medium, to use the biomass in the production of bioethanol. The authors also determined as specific objectives of the present study, to carry out the cultivation of microalgae in an open reactor applying the best condition already seen on the shelf; to produce bioethanol by submerged fermentation from the sugars obtained by hydrolysis; analyze alcohol content, reducing sugars, Brix and concentration; and compare the efficiency of the sugar break in the formation of alcohol.
First, we would like to thank you for your valuable comments and suggestions, which helped us to improve the manuscript. Below we try to address all the points, which you have indicated in your revision. The new/changed text is tracked in the manuscript.
Point 1: The abstract needs a major overhaul. In the beginning there are too many generalities that do not relate to the content of the manuscript. Too much space is devoted to the methodological assumptions and little to the results obtained, the most important discoveries and achievements. These proportions must be changed.
Response 1: Microalgae have a high growth rate, high CO2 absorption capacity, and high content of chlorophyll, proteins, vitamins, mineral salts, carbohydrates, antioxidant substances, and fatty acids. In recent years, Chlorella vulgaris has been widely used as a feedstock for producing third-generation biofuels, how bioethanol. Thus, this work aims to develop a strategy to increase the production scale of the microalgae Chlorella vulgaris grown in distilled reused water, supplemented with a modified BG-11 medium, to use biomass in the production of bioethanol. The total cultivation of 72L presented a concentration of 0.415 g.L-1d-1, with 61.32g of final biomass. To improve carbohydrate extraction, the biomass was pre-treated with sulfuric acid at different concentrations (1.5% and 3% v/v). The hydrolyzed solution was supplemented with YPD (Yeast extract-Peptone-Dextrose) medium and inoculated with Saccharomyces cerevisae yeast, initiating fermentation. In each sample, the Brix degree, cell concentration, reducing sugar concentration, and alcohol content were analyzed. The sample pre-treated with sulfuric acid 1.5% v/v was the one that presented the best result, with alcohol content, after distillation, of 68 oGL (Gay Lussac). It appears that the cultivation of the microalgae Chlorella vulgaris, in scale-up, with reused water, has high potential in the production of third-generation biofuel.
Adjustments were made in lines 13-28.
Point 2: Add more keywords that characterize the content of the manuscript.
Response 2: Open reactor; cultivation; Chlorella vulgaris, carbohydrates; bioethanol. Adjustments were made in line 29-30.
Point 3: The introduction section should precisely present what the research brings to the commonly known knowledge, what was the inspiration for the works, what research hypotheses they wanted to verify, what is the progress of these studies in relation to the ones carried out so far.
Response 3: Adjustments were made in lines 51-63, 84-89, 95
Point 4: There is no clearly and specifically formulated research objective, which should be clear and distinct at the end of the Introduction chapter.
Response 4: This work aims to optimize the cultivation of the microalgae Chlorella vulgaris using a grown in distilled reused water, supplemented with modified BG-11 medium, to use the biomass in the production of bioethanol. The specific objectives of the present study are to: (i) Perform cultivation of microalgae in open reactor applying the best condition previously seen on shelf; (ii) Produce bioethanol by submerged fermentation from the sugars obtained from hydrolysis; (iii) Analyse alcohol content, reducing sugars, Brix and concentration; (iv) Compare the efficiency of sugar breakdown in alcohol formation. Adjustments were made in lines 137 - 143.
Point 5: Please enter the Experiment design or Research organization subchapter in Methodology, where exactly, step by step, the division and sequence of experimental works will be presented. Please consider the block diagram.
Response 5: Adjustments were made in lines 51-63, 84-89, 95.
Point 6: Chapters: Fermentation, Characterisation, Distillation should be corrected with precise names, because it is not entirely clear what the mentioned processes are about.
Response 6: Adjustments were made in lines 170, 186 and 219.
Point 7: Please describe what methods and tests of statistical data analysis were used. A well-conducted statistical analysis of the results is a necessary condition for the credibility of the presented research and formulated conclusions.
Response 7: The analysis referring to figures 2, 3 and 4 was performed with only a single replication, in view of being a base study for bioethanol production from microalgae cultivation. We plan to carry out future trials with replicas for each analysis.
Point 8: Figure 1. Growth kinetics curve over the 28 days of cultivation, whose growth peak occurred 99 at 14 days and the description should be moved to section 3. Results
Response 8: Adjustments were made in lines 232-241.
Point 9: The results should be presented not only in terms of average values, but also with standard deviations. This should be completed in charts, tables and content.
Response 9: As mentioned earlier, the work was a pilot evaluation, whose results showed promising values for suitability of future complementary works.
Point 10: Discussion of the results and discussion is very poor and laconic, it requires deep supplementation and development.
Response 10: Adjustments were made during the discussion.
Round 2
Reviewer 1 Report
After the revisons made by the authors the whole manuscript has been improved and now it is more readable
On the other hand, the authors should take care of the new parts added. There are still some small inacurracies that can be easilty fixed (name of microalgae in old and new parts have to be reported in italics, the name of journals in the reference list have to be reported as Bioresource Technology instead of Bioresource technology for instance, the new units of measure added have to be uniformed as the old one, i.e. g L-1 instead of g/L)
The English is acceptable
Author Response
Response to Reviewer 1 Comments
Review – Round 2
In this manuscript, the authors investigate the process of bioethanol production from the microalgae Chlorella vulgaris in open reactors using distiller water. The main objective was the production of bioethanol to optimize the cultivation of the microalgae Chlorella vulgaris using culture in distilled water of reuse, supplemented with modified BG-11 medium, to use the biomass in the production of bioethanol. The authors also determined as specific objectives of the present study, to carry out the cultivation of microalgae in an open reactor applying the best condition already seen on the shelf; to produce bioethanol by submerged fermentation from the sugars obtained by hydrolysis; analyze alcohol content, reducing sugars, Brix and concentration; and compare the efficiency of the sugar break in the formation of alcohol.
First, we would like to thank you for your valuable comments and suggestions that helped us improve the manuscript. We address all the points indicated in your revision below. The text changes are tracked in the manuscript.
Point 1: After the revisions made by the authors the whole manuscript has been improved and now it is more readable
On the other hand, the authors should take care of the new parts added. There are still some small inacurracies that can be easilty fixed (name of microalgae in old and new parts have to be reported in italics, the name of journals in the reference list have to be reported as Bioresource Technology instead of Bioresource technology for instance, the new units of measure added have to be uniformed as the old one, i.e. g L-1 instead of g/L)
Response 1: The small inaccuracies were adjusted.
Adjustments were made in lines 56, 126, 128, 154, 172-173, 194, 201, 223-224, 248, 270-272, 327, 328.
The references were adjusted as well.

Reviewer 2 Report
Thank you.
Author Response
First, we would like to thank you for your valuable comments and suggestions that helped us improve the manuscript. We address all the points indicated in your revision below. The text changes are tracked in the manuscript.

Reviewer 3 Report
I thank the authors for improving the manuscript and answering the questions in the review. Unfortunately, conducting the research in one repetition completely eliminates their scientific value. The manuscript may be published, however, the presented values must result from at least three repetitions, and the mean-necessary values must be supplemented with standard deviation. This shows the extent of the possible range of volatility.
Author Response
In this manuscript, the authors investigate the process of bioethanol production from the microalgae Chlorella vulgaris in open reactors using distiller water. The main objective was the production of bioethanol to optimize the cultivation of the microalgae Chlorella vulgaris using culture in distilled water of reuse, supplemented with modified BG-11 medium, to use the biomass in the production of bioethanol. The authors also determined as specific objectives of the present study, to carry out the cultivation of microalgae in an open reactor applying the best condition already seen on the shelf; to produce bioethanol by submerged fermentation from the sugars obtained by hydrolysis; analyze alcohol content, reducing sugars, Brix and concentration; and compare the efficiency of the sugar break in the formation of alcohol.
First, we would like to thank you for your valuable comments and suggestions, which helped us to improve the manuscript. Below we try to address all the points, which you have indicated in your revision. The new/changed text is tracked in the manuscript.
Point 1: Unfortunately, conducting the research in one repetition completely eliminates their scientific value. The manuscript may be published, however, the presented values must result from at least three repetitions, and the mean-necessary values must be supplemented with standard deviation. This shows the extent of the possible range of volatility.
Response 1: We appreciate the concern with the repeatability of the data and therefore from the first review and decided to repeat the tests and obtain the values for the calculations of the standard deviations; as can be observed in figures 5, 6, and 7 present in the article. Below are the tables with their respective values.

Round 3
Reviewer 3 Report
Manuscript can be published in current form.